# Contrasting Roles of Ethylene Response Factors in Pathogen Response and Ripening in Fleshy Fruit

**DOI:** 10.3390/cells11162484

**Published:** 2022-08-10

**Authors:** Shan Li, Pan Wu, Xiaofen Yu, Jinping Cao, Xia Chen, Lei Gao, Kunsong Chen, Donald Grierson

**Affiliations:** 1Key Laboratory of Plant Germplasm Enhancement and Specialty Agriculture, Wuhan Botanical Garden, Chinese Academy of Sciences, Wuhan 430074, China; 2College of Agriculture and Biotechnology, Zhejiang University, Zijinggang Campus, Hangzhou 310058, China; 3College of Food Science and Engineering, Hainan University, Haikou 570228, China; 4Zhejiang Provincial Key Laboratory of Horticultural Plant Integrative Biology, Zhejiang University, Zijinggang Campus, Hangzhou 310058, China; 5Plant and Crop Sciences Division, School of Biosciences, University of Nottingham, Sutton Bonington Campus, Loughborough LE12 5RD, UK

**Keywords:** fruit ripening, fruit pathogen response, ethylene response factor (ERF), transcription factor (TF)

## Abstract

Fleshy fruits are generally hard and unpalatable when unripe; however, as they mature, their quality is transformed by the complex and dynamic genetic and biochemical process of ripening, which affects all cell compartments. Ripening fruits are enriched with nutrients such as acids, sugars, vitamins, attractive volatiles and pigments and develop a pleasant taste and texture and become attractive to eat. Ripening also increases sensitivity to pathogens, and this presents a crucial problem for fruit postharvest transport and storage: how to enhance pathogen resistance while maintaining ripening quality. Fruit development and ripening involve many changes in gene expression regulated by transcription factors (TFs), some of which respond to hormones such as auxin, abscisic acid (ABA) and ethylene. Ethylene response factor (ERF) TFs regulate both fruit ripening and resistance to pathogen stresses. Different ERFs regulate fruit ripening and/or pathogen responses in both fleshy climacteric and non-climacteric fruits and function cooperatively or independently of other TFs. In this review, we summarize the current status of studies on ERFs that regulate fruit ripening and responses to infection by several fungal pathogens, including a systematic ERF transcriptome analysis of fungal grey mould infection of tomato caused by *Botrytis cinerea*. This deepening understanding of the function of ERFs in fruit ripening and pathogen responses may identify novel approaches for engineering transcriptional regulation to improve fruit quality and pathogen resistance.

## 1. Introduction

The changes that occur to fleshy fruits during ripening make them more sensitive than unripe fruit to infection by bacteria and fungi [1,2]. The ripening changes are caused by coordinated induction or the repression of multiple genes, influenced by environmental factors such as light, temperature, humidity [3], and internal elements such as phytohormones [4,5], transcription factors (TFs) [5,6] and epigenetic modifications [7,8,9]. The main TFs in tomato (*Solanum lycorpersicum*) involved include MADS, particularly RIN (MADS-RIN), several different NACs including NOR and NOR-like1, CNR and other TFs responsive to auxin, ABA and ethylene [5]. Some TFs, such as ERFs, trigger gene expression and fruit ripening downstream of RIN, CNR and NOR, or act synergistically with these and other TFs to repress or activate different facets of ripening. TFs also influence infection of fruit by fungal pathogens [1,10]. These include a GRAS TF SlFSR [11], SlMYB75 [12], bHLH TF MYC2 [13,14] and an AP2 TF SlSHN3 [15]. Other TFs involved in fruit development, but not directly related to ripening, can also impact fruit susceptibility to *Botrytis cinerea* (*B. cinerea*), such as the KNOX TFs (SlTKN4 and SlTKN2), Golden2-like (SlGLK2) and TFs and SlAPRR2-like, which regulates chlorophyll accumulation and other aspects of chloroplast development before ripening onset [16,17,18,19]. SlGLK2 over-expression has been suggested to favour infections of unripe fruit by *B. cinerea* because of their increased sugar content [10]. Tomatoes engineered to express TFs that upregulate flavonoid biosynthesis genes show an extended shelf life [20,21,22], and this has been attributed to their higher levels of flavonoids, which are important antioxidants. For example, purple Del/Ros1 tomatoes are firmer than WT fruit, and their shelf life was doubled [20]. This is consistent with the ability of flavonoids to scavenge hydroxyl radicals, thus reducing their effects on cell-wall integrity, but other mechanisms may also be involved.

Fungal diseases such as those caused by *B. cinerea* and *Colletotrichum gloeosporioides,* are the most widespread cause of fruit rotting [23]. *B. cinerea* is a well-researched airborne necrotrophic plant pathogen, which can infect more than 200 crop hosts throughout the world [14,24], leading to annual economic losses exceeding USD 10 billion worldwide [25]. *C. gloeosporioides* causes anthracnose, a major disease of many tropical fruits such as avocado (*Persea americana* Mill.), mango (*Mangifera indica* L.), papaya (*Carica papaya* L.) and strawberry (*Fragaria ananassa* Duch.). While the host fruit are still unripe, *C. gloeosporioides* spores produce an infection peg, which penetrates the cuticle of unripe fruit and then becomes quiescent. Growth is only resumed when the fruit begin to ripen and the quiescent fungus is stimulated to produce hyphae and initiate necrotrophic growth, causing the anthracnose fruit-rot symptoms. It is possible that the fungus perceives ethylene or monitors other signals produced by the fruit during ripening or wounding [26], but this requires further investigation. The quiescent phase of *C. gloeosporioides* contrasts with the behaviour of *B. cinerea*, which can infect both unripe and ripe fruits through wounds, causing grey mould. Alkan et al. [27] carried out simultaneous expression analysis of the *C. gloeosporioides* pathogen and fruit during different stages of infection in tomato and showed that over 3000 transcripts were significantly up-regulated, compared to control fruit. The upregulated genes were involved in processes such as the hypersensitive response (HR), phenylalanine biosynthesis, flavonoid biosynthesis and cell wall modification (xyloglucan endotransglucosylases and expansin-like proteins). Several responses were involved in promotion of cell death, including enhanced salicylic acid (SA) signaling, which promotes the respiratory burst, involving nicotinamide adenine dinucleotide phosphate (NADPH) oxidase transcripts, and repression of the jasmonic acid (JA) and ethylene response pathways. Hong et al. [28] characterized the transcriptome of mango infected by *C. gloeosporioides* and identified transcripts for defense-related pathways and other proteins involved in signaling pathways. Genes encoding 373 TFs including MYBs (10.7%-), WRKYs (8.0%-), and ERFs (16.1% of the total TFs) showed increased expression and 13 *ERF* unigenes were up-regulated 1.5- to 85-fold in infected fruits.

The challenge for fruit production is to balance fruit ripening and pathogen resistance to maintain postharvest fruit quality. Understanding the underlying processes of ripening control and the pathogen response, both of which involve ERFs, may provide novel approaches for engineering improvements to fruit quality and pathogen resistance. Ethylene response factors (ERFs) belong to the APETALA2/ethylene response factors (AP2/ERFs) TF family, one of the largest of the 58 plant TF families, defined by their AP2/ERF domains that consist of approximately 60 to 70 amino acids [29]. ERFs act downstream of ethylene signaling [30,31] and play important roles in various aspects of fruit ripening [30,32,33]. ERFs have also been widely reported to be involved in responses to multiple biotic and abiotic stresses in various plant species [34]. In addition to their role in ripening, multiple members of the AP2/ERFs family play a role in fleshy fruit resistance to pathogens [35,36].

In this article, we review and summarize the current status of studies on ERFs that regulate fleshy fruit ripening and fruit resistance to pathogens, as part of an investigation into novel approaches for altering transcriptional regulation to improve fruit quality and pathogen resistance.

## 2. ERF Transcription Factors in Fruit Ripening

Fleshy fruits are classified as climacteric (e.g., apples (*Malus pumila* Mill.), pears (*Pyrus* spp.), bananas (*Musa* spp.), melons (*Cucumis sativus* L.) and tomato) or non-climacteric (e.g., pineapple (*Ananas comosus* L.), strawberry, citrus (*Citrus reticulata* Blanco.)) types. Tomato is a typical climacteric fruit which shows a characteristic rise in respiration (the respiratory climacteric) and ethylene production at the onset of ripening. Ethylene is required for ripening initiation in climacteric fruits and is a major cue that initiates most aspects of ripening [5,32,37], although signaling by other hormones also augments ripening. In the non-climacteric strawberry, however, ABA appears to be the main initiator of ripening [38].

Increases in ethylene production in plants involve changes in expression of multiple ACC synthase (ACS) and ACC oxidase (ACO) enzymes (Figure 1, [4]). Prior to ripening, the basal level of ethylene production (System 1 ethylene) in tomato is approximately 0.05 nL^−1^·g^−1^·h^−1^. This typically rises 100- to 300-fold, and peak ethylene production ranges from 5 to 15 nL^−1^·g^−1^·h^−1^. This increase is due to autocatalytic (System 2) ethylene biosynthesis [39]. Different isoforms of ACS and ACO with different structures and regulatory properties encoded by genes with different transcriptional regulation are involved in system 1 and system 2 ethylene synthesis. Ethylene controls different steps in the ripening process by activating expression of ERFs and other TFs, which upregulate different genes involved in color, flavor, texture and aroma (Figure 1) by individual ERFs that regulate them [4,32,33,40]. Ethylene effectively promotes ripening when applied to mature fruit, and its action can be specifically inhibited by the volatile 1-methylcyclopropene 1-carboxylic acid (1-MCP) [41,42,43], which provides a useful experimental switch and a means of controlling postharvest behaviour of many fruits and flowers.

ERFs are downstream ethylene signaling components and they play important roles in ethylene-dependent developmental processes such as abscission, senescence and ripening of climacteric fruit [30,32,33,44]. The large number of ERFs is believed to explain the diversity and specificity of ethylene responses in plants. Based on distinctive amino acid residues and utilizing the classification adopted for Arabidopsis, at least 81 ERF members were identified from the tomato genome, each of them named with a letter (A–J), plus the subclass and a number [30,33,44]. Many of these ERFs activate the expression of specific genes or groups of genes. The ERF.F subfamily members, however, act as transcriptional repressors, not activators, of downstream target genes and have a characteristic EAR (Ethylene-responsive element binding factor-associated amphiphilic repression) motif (LxLxL and DLNxxP), associated with transcriptional repression [45].

A heatmap shows the different ripening-related expression patterns of the 23 main *ERF* genes expressed during tomato ripening, extracted from publicly available transcriptomes using the online TomExpress platform (http://gbf.toulouse.inra.fr/tomexpress, accessed on 15 March 2022) (Figure 2). Twelve genes (*ERF.A3*, *D3*, *E1*, *E2*, *E3*, *F3*, *F4*, *F5*, *F6*, *H10*, *H12*, *H14*) display an increase in their expression at the ripening initiation stage (Breaker) and reach a peak at the pink or fully ripe stage; one gene (*ERF.F2*) displays an increase in transcripts at the mature green (MG) stage and remains relatively highly expressed during ripening. Six genes (*ERF.B3*, *C1*, *C6*, *D1*, *D2*, *H17*) show an increase at a specific ripening stage (pink or fully ripe) but show no significant changes when ripening is initiated. The other genes (*ERF.B1*, *B2*, *E4*, *F1*) show obvious increases and peak transcript levels at the breaker stage, which supports their potential functions in ripening initiation (Figure 2). It should be noted that 17 of the selected 23 genes, excluding *ERF.C6*, *D1*, *D3*, *E3*, *H14*, *H17*, were also considered as the best candidates for involvement in ripening initiation and progression, based on their high expression levels and ripening-related pattern of transcript accumulation [33].

Previous studies recognized that ERF.E4 (alternatively named SlERF6 [47]; or named LeERF1 [48]) and ERF.B3 [49] TFs play key roles in ethylene biosynthesis, signaling and ripening. Transcripts of *ERF.F5* (alternatively named *LeERF3*/*LeERF3b*), a repressor class *ERF* gene that does not respond to abiotic stresses or wounding, accumulate before ripening onset and decline sharply thereafter [50]. *ERF.F4* (alternatively named *ERF9*) transcripts also increase significantly late in tomato fruit developmental and their transcript accumulation is negatively correlated with flavonoid biosynthesis [51]. Several more *ERF* genes appear to be involved in tomato fruit ripening, such as *SlPti4* (also named *ERF.A3*) [52] and *ERF.F12* [30].

Other climacteric fruits, in addition to tomato, also undergo ethylene synthesis and evolution at the onset of ripening and multiple members of the AP2/ERFs family appear to have ripening-related functions in other fruits (discussed later). Ethylene signaling via ERFs is transmitted to the promoters of genes such as *ACC oxidase* (*ACO*), *ACC synthase (ACS)*, *polygalacturonase* (*PG*) and *phytosynthase* (*PSY*), etc., which are involved in multiple ripening pathways. In peach (*Prunus persica* L.), ABA is also involved in ripening regulation, and PpERF3 activates and PpeERF2 suppresses ABA biosynthesis by, respectively, positively and negatively regulating transcription of 9-cis-epoxycarotenoid dehydrogenase (NCED) genes *PpeNCED2/3,* involved in ABA biosynthesis [53,54]. PpERF/ABR1 bind directly to the promoter of cell-wall gene *PpPG* and activate its expression in peach fruit [55]. In durian (*Durio zibethinus* L.), DzERF6 and DzERF9 have been recognized as the repressor and activator, respectively, of ethylene biosynthesis [56]. The banana (*Musa acuminata* AAA group, cv. Cavendish) MaERF9, apple (*Malus domestica* Borkh) MdERF1, and apple (*Malus domestica* Golden Delicious) MdERF3 are positive activators [57,58,59], whereas MaERF11 (Cavendish) and MdERF2 (Golden Delicious) repress fruit ripening [59,60]. MaERF9 and MaERF11 regulate the transcript levels of the ethylene biosynthetic genes *ACO1* and *ACS1* by binding to their promoters, and they have also been reported to physically interact with ACO1 protein [57]. MaERF11 also interacts with a histone deacetylase MaHDA1, forming a TF complex that represses genes such as *MaACO1* and *expansins*, in a process involving histone deacetylation [60]. In apple (*Malus domestica*), MdERF2 is involved in three interactions regulating *MdACS* expression. The repressor MdERF2 and activator MdERF3 regulate the transcript level by binding to the promoters of *MdACS* genes. MdERF2 also inhibits *MdERF3* activity by binding to the dehydration-responsive (DRE) element (a core sequence of CCGAC) in the promoter, indirectly suppressing *MdACS* expression. Additionally, a direct interaction between MdERF2 and MdERF3 modulates the binding of MdERF3 to the *MdACS* promoter, suppressing its expression in fruit flesh [59]. *MdPSY1* and *MdPSY2* are involved in carotenoid accumulation in apple (Royal Gala) fruits and are activated by AP2/ERF TFs such as AP2D15, AP2D21, AP2D26 and MdAP2-34 [61,62].

ERFs have also been reported to function in non-climacteric fruit ripening and development of quality attributes. This offers some support for the suggestion that ethylene signaling is required for the process, although another interpretation is that some ERF-like genes are actually regulated by other hormones. CitERF16 activates *CitSWEET11d* expression in citrus (*Citrus unshiu*), which promotes sucrose accumulation in both citrus and tomato fruit [63]. *CitAP2.10* is involved in regulating the synthesis of (+)-valencene in Newhall orange (*Citrus sinensis*), by upregulating the transcription of *terpene synthase 1* (*CsTPS1*) [64]; CitERF71 physically binds to the *CsTPS16* promoter and contributes to *E*-geraniol synthesis in sweet orange (*C. sinensis* Osbeck) fruit [65]. In strawberry, FaERF#9 interacts with FaMYB98 to form an ERF-MYB complex, which activates the quinone oxidoreductase (*FaQR*) promoter. FaQR catalyses the last step in the 4-hydroxy-2,5-dimethyl-3(2*H*)-furanone biosynthesis pathway, which makes a major contribution to strawberry fruit aroma [66]. In ripening watermelon (*Citrullus lanatus*), the accumulation of *ClERF069* transcripts is negatively correlated with the sugar and lycopene content, and its overexpression in tomato delayed the ripening process, reducing ethylene production and the accumulation of lycopene and *β*-carotene, confirming that ClERF069 negatively regulates fruit ripening [67]. Thus, there is substantial evidence for involvement of ERFs in regulating aspects of ripening in climacteric and non-climacteric fruits. However, it should be emphasized, as mentioned above, that ERFs have been identified mainly by gene homology, and it is possible that in non-climacteric fruit they are regulated by other hormones, in addition to ethylene, such as ABA [38].

## 3. Fruit Responses to Pathogen Infection

Ripe fruit are much more sensitive to pathogen infection then unripe ones, as shown by taking the climacteric fruit tomato as an example. First, tomato pleiotropic ripening mutants, e.g., *rin*, *nor*, *Nr*, are unable to ripen and do not rot for many months. The senescence process that normally begins after ripening greatly increases the susceptibility of fruits to pathogenesis and rotting. Alterations in fruit texture, increases in the accumulation of excess reactive oxygen species (ROS), increases in cell membrane permeability and metabolic disorder lead to irreversible damage, cell disintegration and increased sensitivity to fungal infection [68,69,70]. The cell-wall degradation that occurs during ripening, causing softening, is due to the upregulation of a battery of cell-wall-modifying enzymes. The action of ROS also enhances softening [71] and, together with other changes in transcript-encoding cell-wall enzymes, increases the susceptibility to pathogen infection. Second, ripening enhances the likelihood of damage to fruit, which can generate wounds for pathogen entry [10,27]. For example, genes encoding proteins involved in cell-wall changes, such as expansin (Exp1), pectinmethylesterase (PME), polygalacturonase (PG) and pectate lyase (PL), all contribute to a reduction in the firmness of tomato fruit, which increases the susceptibility to pathogenic fungi [70,72,73]. Third, the accumulation of metabolites such as amino acids, organic acids, sugars, and some secondary metabolites provides a rich source of nutrients for colonizing pathogens. Alterations in intra- and extra-cellular pH, ROS levels and increases in lipid peroxidation and protein oxidation also change the cellular environment during ripening and provide energy and signals for pathogens, generating a favorable environment for growth and reproduction for invading fungi [10]. ROS is not only important because of its effects on plant metabolism but also because the increase in ROS that occurs in response to pathogenic fungal initiates the host plant oxidative burst [10].

### 3.1. General Responses to Infection and the Involvement of ERFs

Fruit pathogens include pathogenic fungi, bacteria, viruses, viroids, etc. [74]. Compared to parasitic microoganisms, pathogenic fungi with necrotrophic lifestyles are much more devastating, killing the host tissue and eventually causing rotting. Necrotrophic fungi include *B. cinerea*, *Rhizopus stolonifera*, *Fusarium acuminatum*, *C. gloeosporioides*, and plant immune responses against these fungi involve pathogen-associated molecular pattern (PAMP) recognition and mitogen-activated protein kinase (MAPK) cascades. Phytohormones such as ethylene, JA and SA also coordinate and regulate the production of multiple complex stress-related secondary metabolites [73].

Active defenses switched on when plants are infected by pathogens include an oxidative burst, expression of defense-related genes, such as *pathogenesis-related* (*PR*) genes, and the accumulation of antimicrobial compounds [75,76,77]. *PR* genes are regulated by several plant hormones, such as SA and JA [78]. TFs are central components of plant defense signaling and adaptation mechanisms [74]. WRKY62 and WRKY 70 TFs are involved in the JA and SA signaling pathways [79,80], and WRKY TFs can regulate the transcription of many *PR* genes by binding to the W-box in their promoters [81]. In addition to WRKYs, ERFs have been widely reported as being involved in regulating ripening and responses to various biotic and abiotic stresses [34] by binding to a cis-acting element, the AGCCGCC (the GCC box) [34] element, which is highly enriched in promoter regions of multiple genes expressed in response to pathogen infection. There are several examples in Arabidopsis of both positive- and negative-acting ERFs responding to infection (see Amorim et al. [82]), and the ERF repressors generally have the EAR motif. For example, ERF1 is known to up-regulate defense genes, leading to enhanced resistance in response to several fungi [83]. In contrast, other ERFs, such as ERF9 and ERF14, inhibit the expression of *PR-1* during infection by *Piriformospora indica*, and ERF9 negatively regulates defense gene expression mediated by ET/JA signaling in response to *B. cinerea* infection [84].

### 3.2. ERF Factors Induced in Fruit Responses to B. cinerea

Tomato is the most important global vegetable crop. Fungal phytopathogens cause widespread losses and also reduce the ripe fruit quality. Among all the fungal pathogens, the grey mould disease caused by *B. cinerea* causes one of the biggest problems, and the tomato-*B. cinerea* pathosystem has become a model for studying quantitative host–pathogen interactions [85]. In order to analyze the function of ERFs in fruit responses to *B. cinerea* infection, we carried out comprehensive transcriptomic profiling of tomato *ERF* genes in WT fruits among AC_healthy, AC_wounded (wounded but not inoculated), AC_*B. cinerea* (wounded with *B. cinerea* inoculation) at two different ripening stages mature green (MG) and red ripe (RR)) using information from the NCBI database. A heatmap representing their expression pattern shows 33 genes, excluding those with low levels of transcripts (selected by FPKM > 10) (Figure 3a).

These 33 genes have different pathogen-response-related patterns; 16 genes (*ERF.A1*, *A4*, *B2*, *B12*, *B13*, *C1*, *C3*, *C4*, *C6*, *D2*, *D6*, *D7*, *F4*, *F5*, *G2*, *H9*) display an obvious increase (fold change above 2.0) in their expression after *B. cinerea* inoculation compared to wounding alone at either or both ripening stages (MG and RR), and seven of them (*ERF.A1*, *A4*, *B12*, *C3*, *C4*, *C6*, *H9*) have strongly increased transcript levels (fold change above 10). Transcripts of *ERF.A1* and *C6* were strongly induced at both MG and RR ripening stages; the other genes (*ERF.A3*, *B1*, *B3*, *E1*, *E2*, *E3*, *E4*, *E5*, *F1*, *F2*, *F3*, *F6*, *H7*, *H10*, *H12*, *H14*, *H16*) showed no obvious increased transcript levels after wounding or *B. cinerea* inoculation.

Multiple ERF TFs are involved in tomato fruit resistance to *B. cinera* (Table 1). Expressions of *ERF.A2* (*ERF1*) and *ERF.F5* (*ERF3*) were upregulated in fruit after *B. cinerea* infection at the MG and RR stage, respectively [86]. ERF.A3, B2 (ERF5), C3, C6 and G2 regulate the expression of genes including *Pathogenesis-Related* (*PR1-5*), *Plant Defensin* (*PDF1.2*), the pathogenesis response marker gene *PR-STH2* (formerly pathogen-activated gene of potato and the corresponding protein) and *Thionin* (*Thi2.1*) involved in *B. cinerea* infection [87,88,89,90]. ERF.A3 and F12 regulate the transcription of other TF genes including *Related-to-ABI3/VP1* (*RAV1*), *MYC2*, *cytokinin*
*response*
*factor*
*1* (*SlCRF1**)*, *tomato stress-responsive factor 1* (*TSRF1*), *Pti5* and *Pti6* involved in *B.cinerea* resistance [30,87,90,91,92].

The tomato *SlERF2* (*ERF.E1*) gene participates in the disease resistance response and ERF2 overexpression enhanced tomato fruit resistance against *B. cinerea*. Methyl jasmonate (MeJA) is involved in this process and supplying it externally increased ethylene production, chitinase, β-1,3-glucanase, peroxidase and phenylalanine ammonia-lyase activities, and increased the PR proteins and phenolic content. Furthermore, MeJA enhanced the disease response in *ERF2*-overexpressing tomato fruit, whereas the effect was reduced in antisense *SlERF2* tomato fruit [35]. The overexpression of *ERF.B2* (*SlERF5*) and *ERF.F5* (*SlERF3b*) plays an important role in the immune response to *B. cinerea* by upregulating the JA/ET signaling pathways [94]. *B. cinerea* infection leads to the upregulation of *SlERF.A1, SlERF.A3, SlERF.B4*, and *SlERF.C3* defense signaling hormones such as SA, MeJA, and 1-aminocyclopropane-1-carboxylic acid (ACC, an ethylene precursor). The inhibition of either *SlERF.B1* or *SlERF.C2* by virus-induced gene silencing (VIGS)-proved lethal and silencing of *SlERF.A3* (*Pit4*) significantly reduced vegetative growth of tomato plants. Silencing of *SlERF.A1, SlERF.A3, SlERF.B4*, or *SlERF.C3* increased the susceptibility to *B. cinerea* and reduced accumulation of jasmonic acid and ethylene-responsive defense genes and H_2_O_2_ accumulation, which is involved in the plant hypersensitivity response. In response to *B. cinerea*, *SlERF.A3* silencing also reduced the resistance to *Pseudomonas syringae* pv. *tomato* (*Pst*) DC3000, whereas silencing of *SlERF.A1, SlERF.B4* or *SlERF.C3* had no effect on resistance to this bacterium [36].

### 3.3. ERFs Interact with Other Ripening Regulators and Affect Fruit Resistance to B. cinerea Infection

A typical sign of tomato fruit ripening is the burst of ethylene production, which is essential for ripening initiation and progression [32,37]. Compared to unripe fruit, ripe fleshy fruit are more sensitive to mould infection [2], and this implies, as discussed above, that changes occurring during ripening cause fruit to become more susceptible to attack by pathogens. Several ripening-related mutants of the model climacteric fruit tomato [101], including *ripening inhibitor* (*rin*) [102], *non-ripening* (*nor*) [103], *colorless non-ripening* (*cnr*) [104] and *Never-ripe* (*Nr*) [105], make it possible to dissect and analyze the network governing gene expression changes during ripening and fruit pathogen infection.

It has been reported that mutant *nor* and *rin* fruit have similar disrupted ripening phenotypes and severely inhibited ethylene production [102,103], while the *Nr* fruit have disrupted ethylene signaling, which inhibits the ripening phenotype, and *Cnr* fruit remain partially unripe because of genome DNA methylation [106]. Tomato fruit susceptibility to *B. cinerea* infection is determined by disease incidence and disease severity (lesion growth, in mm) after inoculation. It was reported that MG-like fruit of *Cnr* was the only unripe fruit susceptible to *B. cinerea* infection, and *Cnr* RR-like showed greater susceptibility than wildtype (WT) Ailsa Craig (AC) fruit. Both MG-like and RR-like stages of *rin* fruit showed similar or slightly reduced susceptibility compared to WT AC fruit [73], and *Nr* fruit were fully resistant at the MG-like stage but only partially resistant at the RR-like stage [1].

However, ethylene production was induced by *B. cinerea* infection in WT, *rin* and *Nr* tomato fruit, which was shown by the fold change of ethylene production comparing inoculation to wounded fruits at 3 days post-inoculation (dpi), at the MG stage of AC (1.3-fold), *rin* (15.5-fold), *nor* (4.4-fold), *Nr* (22.5-fold) fruit and RR-like stage of AC (2.6-fold), *rin* (75.0-fold), *nor* (3.0-fold), *Nr* (3.2-fold) fruit [1] (ethylene production in *Cnr* fruit after inoculation was not checked). If ethylene was involved in disease susceptibility or response, in WT AC and *rin* fruit, treatment with 1-MCP might be expected to reduce their susceptibility to infection by pathogens compared to controls, but 1-MCP-treated WT RR fruit were not more resistant [1]. This suggests that ethylene production levels are not directly related to differences in susceptibility or resistance and some factor(s) other than ethylene that change during ripening may promote susceptibility.

To understand more about the role of ERFs in fruit ripening, we compared the accumulation of their transcripts at the MG and RR ripening stages in the extremely sensitive *cnr*, medium sensitive WT AC, slightly resistant *rin* and resistant *nor* ripening mutants. *Nr* was not considered due to its altered ethylene signaling mechanism. In interpreting the results of these experiments, it is important to recognize that it has recently been demonstrated that the *nor* and *rin* mutations produce aberrant TF proteins (either truncated NOR in *nor* or fused RIN-MC in *rin*) that have, through mutation, acquired the ability to negatively regulate some of their gene targets [37,107,108,109,110,111]. Furthermore, *CNR* knock-out fruit displayed only slightly delayed ripening, which is puzzling since it was previously recognized as a master ripening regulator [109]. Functional identification of RIN in fruit resistance to *B. cinerea* has been re-examined using knocking out (KO) technology, where RIN is absent [112], which contrasts with the previous comparison between WT and *rin* mutant, where MADS-RIN is a repressor [1,73]. The expression pattern heatmap for these new results, excluding a low level of transcripts (FPKM > 10), identified 33, 35, 37, 32 genes that had higher level transcripts (selected by FPKM > 10) in WT AC (Figure 3a), *rin* (Figure 3b), *cnr* (Figure 3c) and *nor* (Figure 3d) fruits, respectively, in healthy, wounded and *B. cinerea* infected (wounded plus *B. cinerea*) fruits.

In WT AC fruits, the selected 33 genes have different pathogen-response-related expression patterns. Sixteen of them have obvious increases in response to *B. cinerea* infection, and seven of them display strongly induced transcript levels, discussed above (Figure 3a). The same 35 genes have different pathogen-response-related expression patterns in *rin* fruits; 19 of them (*ERF.A1*, *A3*, *A4*, *A5*, *B12*, *B13*, *C1*, *C3*, *C4*, *C6*, *D2*, *D6*, *D7*, *E1*, *E4*, *F4*, *F5*, *G2*, *H9*) show obvious increases (fold change above 2.0) in response to *B. cinerea* infection and *ERF.A1*, *A4*, *C1*, *C3*, *C4*, *C6* and *ERF.A5*, *B12*, *H9* display much higher transcript levels (fold change above 10.0) at one or both ripening stages (Figure 3b). In *cnr* fruits, 20 of the selected 37 genes, (*ERF.A1*, *A3*, *A4*, *A5*, *B12*, *B13*, *C1*, *C3*, *C4*, *C6*, *C9*, *D1*, *D2*, *D6*, *D7*, *E1*, *E4*, *F4*, *H6*, *H9*) show obvious increases (fold change above 2.0) in response to *B. cinerea* infection, and 2 (*ERF.A5*, *C4*) and 11 (*ERF.A1*, *A4*, *B12*, *B13*, *C3*, *C6*, *C9*, *D1*, *D2*, *H6*, *H9*) of them display strongly induced transcript levels (fold change above 10.0) at one or both ripening stages (Figure 3c). In *nor* fruits, 9 of the selected 32 genes (*ERF.A1*, *C1*, *C3*, *C4*, *C6*, *D2*, *D7*, *E1*, *G2*) showed an obvious increase (fold change above 2.0) in response to *B. cinerea* infection, and 1 (*ERF.C6*) and 3 (*ERF.A1*, *C3*, *C4*) were strongly induced (fold change above 10.0) at both or one ripening stage (Figure 3d). Overall, six genes (*ERF.C1*, *C3*, *C4*, *C6*, *D2*, *D7*) showed obvious increases in all of the four tomato types; transcripts of six genes (*ERF.A4*, *B12*, *B13*, *D6*, *F4*, *H9*) were obviously increased in AC, *rin* and *cnr*, but not in the *B. cinerea*-resistant *nor* mutant fruit, and only *ERF.E1* expression was significantly induced in three ripening mutant but not in WT AC fruit. The susceptibility to *B. cinerea* infection of AC and *rin* mutant fruit was similar, and they share the expression of fifteen genes in common, which displayed obvious upregulated expression patterns in both fruit during the response to infection (Figure 3).

### 3.4. Tomato ERF Factors Function in Fruit Response to Other Pathogens

Gray mold, caused by *B. cinerea,* is a devastating disease, and is an important model for studying plant–necrotroph interactions [10]. Both *Fusarium acuminatum* (*F. acuminatum*) and *Rhizopus stolonifer* (*R. stolonifer*) have also been reported to infect various plant tissues, especially fleshy fruit [113]. *F. acuminatum* is one of the most toxic species of *Fusarium* and induces host cell death and tissue necrosis and by producing strong mycotoxins, such as trichothecene and fumonisins [100]. *R. stolonifer* is able to grow rapidly and is a very destructive postharvest pathogen [114], causing rotting by secreting cell-wall-degrading enzymes.

Comprehensive transcriptomic profiling of tomato *ERF* genes in WT fruits of healthy, uninoculated wounded *F. acuminatum* (wounded with *F. acuminatum* inoculation), *R. stolonifer* (wounded with *R. stolonifer* inoculation) at two different ripening stages (MG and RR) was carried out using information from the NCBI database. A heatmap representing their expression pattern shows 37 *ERF* genes excluding those with a low level of transcripts (selected by FPKM > 10) (Figure 4).

At the MG stage, five genes (*ERF.A1*, *A4*, *B2*, *C1*, *C6*) showed an obvious increase (fold change above 2.0) compared to wounded fruits after *F. acuminatum* inoculation, but there were no obvious changes in levels of transcripts after *R. stolonifer* inoculation (Figure 4). At the RR stage, seven genes (*ERF.A1*, *A4*, *B12*, *C1*, *C6*, *G2*, *H9*) showed an obvious increase (fold change above 2.0) in their expression after *F. acuminatum* inoculation compared to wounded fruit, and three (*ERF.A1*, *A4*, *B12*) showed strong induction of transcript levels (fold change above 10). At the RR stage, after *R. stolonifer* inoculation, 22 genes (*ERF.A1*, *A4*, *A5*, *B2*, *B4*, *B5*, *B12*, *B13*, *C1*, *C4*, *C6*, *D2*, *D4*, *D6*, *E1*, *F1*, *F4*, *F5*, *G2*, *H9*, *H12*, *H14*) displayed an obvious increase (fold change above 2.0) in their expression and nine of them (*ERF.A1*, *A4*, *A5*, *B12*, *B13*, *C4*, *C6*, *D4*, *H9*) were strongly induced (fold change above 10). The others (*ERF.A3*, *B1*, *B3*, *C3*, *D7*, *E2*, *E3*, *E4*, *E5*, *F2*, *F3*, *F6*, *H7*, *H10*, *H16*) showed no obvious increased transcript levels after *F. acuminatum* or *R. stolonifer* inoculation (Figure 4).

Except for above expression patteren related to fungal pathogen response, multiple ERFs also function in resistance to other pathogens, as listed in Figure 5. Overexpression of *ERF1* (*ERF.H1*) in tomato-fruit-enhanced resistance to *Rhizopus nigricans*, and this was associated with a substantial accumulation of transcripts of *PR1a*, *PR5*, *Chi1* and *PAL*. It was concluded that although changes in ethylene production can occur, it does not play a pivotal role in fruit resistance to *R. nigricans* [100]. Tomato *ERF.C1* (also named *SlERF1*, *TERF1 or JERF2*) is thought to be involved in the chitosan (CHT)-induced systemic acquired resistance (SAR) response [95]. *ERF.B3* (also named *LeERF4*) and *ERF.D2* are also induced by infection by *Trichoderma harzianum* (strain T22) and aphid *Macrosiphum euphorbiae* infestation [115] and the *ERF.D2* (also named *ACE43*) gene is required for *N. benthamiana* non-host resistance to *Xanthomonas oryzae* pv. *Oryzae* [116]. *ERF68* (also named *ERF.A1*) silencing has been shown to increase susceptibility of tomatoes to two incompatible *Xanthomonas* spp. and is associated with the altered expression of genes involved in ethylene production and SA, JA and hypersensitive response pathways. Target genes assay by chromatin immunoprecipitation combined with high-throughput sequencing (ChIP-seq) indicated that ERF68 is involved in promoting response to infection, including hypersensitive cell death, by modulating multiple signaling pathways [93].

Tomato SlERF01 (also named ERF.C4) activates the expression of *PR1* and plays a key role in multiple SA, JA and ROS signaling pathways believed to be important for resistance to *S. lycopersici* infection [96]. *ERF2* (also named *ERF.C6*)-silenced tomato plants showed susceptibility after inoculation with *S. lycopersici*, which might be due to a decreased hypersensitive response involving reduced catalase, peroxidase and superoxide dismutase leading to a reduction in ROS production. Furthermore, the results indicated that ERF2 may directly or indirectly regulate *Pto*, *PR1b1* and *PR-P2* expression, believed to be involved in the defense response, thereby also enhancing tomato resistance [98].

The transcriptional activator TSRF1 enhances plant resistance to the bacterial pathogen *P. syringae* and fungal disease agent *B. cinerea*. Overexpressing *TSRF1* (also named *ERF.C4*) in tomato and tobacco activates the expression of *PR* genes, enhancing plant resistance to *R. solanacearum,* the causative agent of bacterial wilt disease [92]. Expression of *TSRF1* is enhanced by ethylene-induced salicylic acid accumulation and activates the expression of *PR1*, *PR2* and *PR3* [97].

*SlERF84* (*ERF.D6*) overexpression in *Arabidopsis* plants, on the other hand, led to more severe symptoms with extensive chlorotic lesions after inoculation with *Pst* DC3000. Several other well-documented genes, such as *AtPR1* and *AtPR3*, have been suggested to play important roles in resistance to pathogens, and their transcript levels were much lower in *ERF.D6*-overexpressign lines compared to controls, after inoculation with *Pst* DC3000 [99]. The miR172-mediated silencing of *AP2d* in *Solanum pimpinellifolium* L3708 conferred greater resistance to *Phytophthora infestans* (*P. infestans*) infection [117].

The monopartite tomato yellow leaf curl virus (TYLCV) of the genus *Begomovirus* in the Geminiviridae family induces serious damage to tomato production and quality due to the serious symptoms, which include leaf yellowing and curling as well as the occurrence of shrunken new leaves and cessation of plant growth [118]. Five different tomato cultivars which show different resistances or susceptibilities to the virus, including Hongbeibei (highly resistant), Zheza-301, Zhefen-702 (both resistant), Jinpeng-1, and Xianke-6 (both susceptible), were selected for gene expression pattern assay. In total, 22 *ERFs* were identified in response to tomato TYLCV using transcriptome data; Two of them (*Soly19* (*ERF.C6*), *Soly36* (*ERF.B2*), *Soly66* (*ERF.B3*), *Soly67* (*ERF.B1*) and *Soly106* (*ERF.C4*)) were selected for further assay and showed different response to TYLCV virus between these cultivars [118].

### 3.5. ERF Factors Function in Other Fleshy Fruits in Response to Pathogens

*ERF* genes play critical roles not only in tomato fruit resistance to pathogens but also in other fruit of both climacteric and non-climacteric (e.g., strawberry, citrus) types, listed in Table 2 and Figure 6.

Interestingly, heat (52 °C for 3 min) reduced lesion sizes in infected banana, and it has been suggested that heat activates defense responses by the up-regulation of *MaERF1* [128]. Li et al. [129] characterized *CpERF1, 2, 3, 4* from papaya that appear to play different roles during fruit development, ripening and responses to stress. *CpERF2* and *CpERF3* are closely associated with fruit ripening, and the accumulation of *CpERF1*, *CpERF3* and *CpERF4* transcripts were induced by the ethylene perception inhibitor 1-MCP. Conversely, *CpERF2* was repressed by 1-MCP and its transcripts accumulated in response to ethylene. *CpERF2* and *CpERF4* are involved in the response of papaya to *C. gloeosporioides* stress, and the accumulation of *CpERF2*, *CpERF3* and *CpERF4* transcripts is rapidly activated by low temperatures, whereas *CpERF1* was induced by high temperature.

In strawberry (*Fragaria vesca*) fruit infected with *B. cinerea*, the expression of *WRI1*, *ERF061*, *ERF053* were significantly upregulated in unripe (white stage) fruit [119]. Transcripts of two genes encoding different ERF2-like and five other genes also encoding ERF5 sequences were induced significantly in fruit (“Sunnyberry” (gray mold-resistant) and “Kingsberry” (gray mold-resistant)) as they transitioned from the immature-green (IG) stage to the mature-red (MR) stage [120]. In apple (*Malus domestica*) ‘Jonagold’ fruit, accumulation of transcripts for all four *MdERFs* (*MdERF3**–6*) was ethylene-dependent and induced by wounding or by *B. cinerea* infection [121]. Noticeably, ripening activator MdERF3 directly binds to *MdACS* genes and represses its expression by recruiting repressor MdERF2 [58,59], which indicates the dual role of MdERF3 in both fruit ripening and pathogen response. Overexpression of *MdERF11* in apple (*Malus domestica* Borkh) callus cells significantly increased their resistance to *B. dothidea* infection, whereas silencing *MdERF11* resulted in reduced resistance. MdERF11 increases plant defense against *B. dothidea* by stimulating the SA biosynthesis pathway [122], and MdERF100 (in *Malus domestica* Gala) interacts directly with MdbHLH92 to mediate resistance to powdery mildew by upregulating the JA and SA signaling pathways [123]. Overexpression of *VvERF1* in grape (*Vitis vinifera* Kyoho) fruits reduced the susceptibility to *B. cinerea* infection [124]. In Chinese wild *Vitis quinquangularis VqERF112*, *VqERF114* and *VqERF072* transcripts increased in response to the powdery mildew pathogen *Pseudomonas syringae* pv. Tomato (*Pst*) DC3000, *B. cinerea* and to treatments with ET, SA, MeJA or ABA hormones. When *VqERF112*, *VqERF114* and *VqERF072* were overexpressed in Arabidopsis, resistance to *Pst* DC3000 and *B. cinerea* increased and the accumulation of transcripts for SA signaling-related genes *AtNPR1* and *AtPR1* and JA/ethylene signaling-related genes *PLANT DEFENSIN 1.2*
*(PDF1.2**)*, *LIPOXYGENASE 3* (*LOX3*), *BASIC CHITINASE* (*PR3*) and *HEVEIN-LIKE* (*PR4*) also increased [125].

Expression profiling of genes such as *VvERF041* and *VvERF069*, *VvERF071*, *VvERF072* and *VvERF099* showed they were induced after infection of a *B. cinerea*-resistant variety, Shuangyou (*V**itis amurensis*) and a susceptible variety, Red Globe (*V**itis vinifera*), indicating that they have a potential role in responding to attack by pathogens [126]. ERF members VpERF1, VpERF2 and VpERF3 in a highly powdery mildew (PM)-resistant Chinese wild *Vitis pseudoreticulata* were associated with disease resistance, and it was demonstrated that *VpERF2* and *VpERF3* over-expression in transgenic tobacco and *VpERF1* overexpression in transgenic *Arabidopsis* led to enhanced resistance to the fungal pathogen *Phytophtora parasitica* var. *nicotianae* Tucker and also the bacterial pathogen *Ralstonia solanacearum* [127].

In citrus (*Citrus sinensis*), the role of *CsAP2-09* was characterized using over-expression and RNAi silencing strategies. The diseased lesions and disease index in transgenic citrus overexpression lines infected with *Xanthomonas citri subsp* (*Xcc*) were significantly decreased while they were significantly enhanced in RNAi lines. Comparison of the transcriptomes of WT and overexpression lines revealed that some of the genes associated with increased expression were involved in phenylpropanoid biosynthesis, pathogen responses, and transcriptional regulation [130].

## 4. Conclusions

Fleshy fruit production and quality formation is severely affected by biotic stress from fungi, bacteria, and viruses, and clarifying plant disease resistance mechanisms is a critical necessity for improving fruit production and quality by breeding. Plant defenses against microbial attack are activated rapidly by signaling pathways that promote cellular and molecular processes that contribute to resistance [131]. These include the accumulation of ROS, ROS signaling, cell-wall remodeling, the activation of defense-related genes and accumulation of antimicrobial compounds. ERFs belong to one of the most important families involved in the activation or inhibition of target genes expressed during ripening and in response to biotic stress, either alone or in conjunction with other TFs. ERFs also play an important role in fruit ripening and quality development, and there are significant differences in the ERFs expressed during ripening and in response to infection.

Of the 81 identified tomato ERF genes, eighteen are recognized as the best candidates for being involved in ripening initiation and progression, although the transcript levels of one (*ERF.F12*) are not very high. Twenty three *ERF* genes (either in WT AC, *rin*, *nor*, *cnr*) respond to *B. cinerea* infection, and eight *ERF* genes also respond to infection by three fungi investigated; five of these *ERF* genes (*ERF.A1*, *A4*, *B12*, *G2*, *H9*) were not involved in ripening initiation or progression, and the other three (*ERF.B2*, *C1*, *C6*) showed a peak expression at a specific ripening stage; twenty-eight *ERF* genes could respond to one or more fungi; five of them (*ERF.A3, E1, F4, F5, H12*) also showed ripening-related expression patterns, and could be recognized as both tomato fruit ripening and fungal resistance regulators.

Thus, ERFs are widely reported to be involved in ripening regulation and responses to fungal, bacterial and virus infection in climacteric (peach, durian, apple, banana, papaya, mango) and non-climacteric fruits (citrus, strawberry, watermelon, grape). Different ERFs regulate aspects of ripening and responses to infection and are involved in the stress-related synthesis of PR proteins, phytohormones ethylene and JA signaling pathways. In tomato, key ERFs regulating ripening, responses to infection, or both have been identified (Figure 5). Further study of the molecular mechanism and the regulatory interactions between these regulators may reveal opportunities for improving fruit resistance and quality.

## Figures and Tables

**Figure 1 cells-11-02484-f001:**
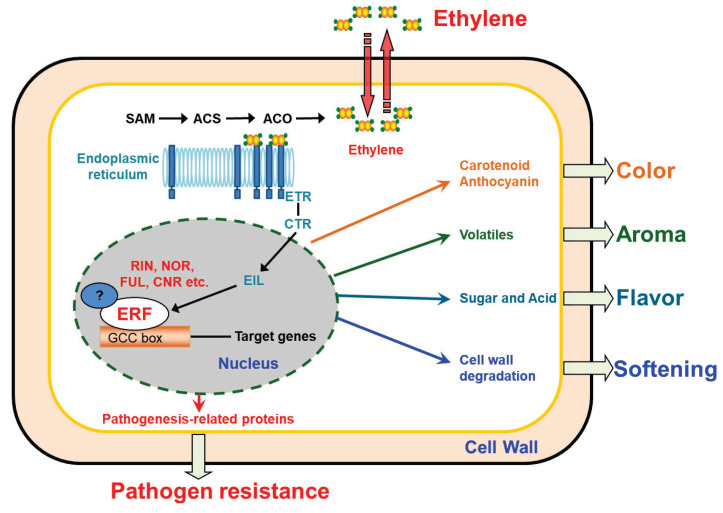
The involvement of ethylene response factor (ERF) in tomato fruit ripening and pathogen response. Ethylene synthesis relies on the activity of ethylene precursor 1-aminocyclopropane-1-carboxylic acid (ACC) synthases (ACS) and ACC oxidases (ACO) which transform S-adenosyl-L-methionine (SAM) into ACC and convert ACC into ethylene, respectively. In tomato, there are 14 ACS (ACS1A, ACS1B, ACS2–13) and 6 ACO (ACO1–6) members; both ACS1A and ACS6 are involved in system 1 ethylene biosynthesis and the autocatalytic ethylene synthesis is catalyzed by ACS2 and ACS4. ACO1 and ACO4 are responsible for the transition to system 2 ethylene because their transcripts accumulate with the climacteric rise of ethylene production [32]. The ERFs function downstream of the ethylene signaling chain (ethylene receptors (ETR), constitutive triple-response (CTR), EIN3-Like (EIL)), and play roles in fruit color, aroma and flavor formation, softening changes and pathogen resistance by regulating different facets of the ripening response by binding to target gene GCC-box elements and transactivation promoters of genes involved in accumulation of carotenoid or anthocyanin and volatiles, sugar and acid metabolism, cell-wall degradation and pathogen response. Ripening regulators such as MADS-RIN (RIN), SBP-CNR (CNR) and NAC-NOR (NOR) also act on expression of multiple downstream target genes.

**Figure 2 cells-11-02484-f002:**
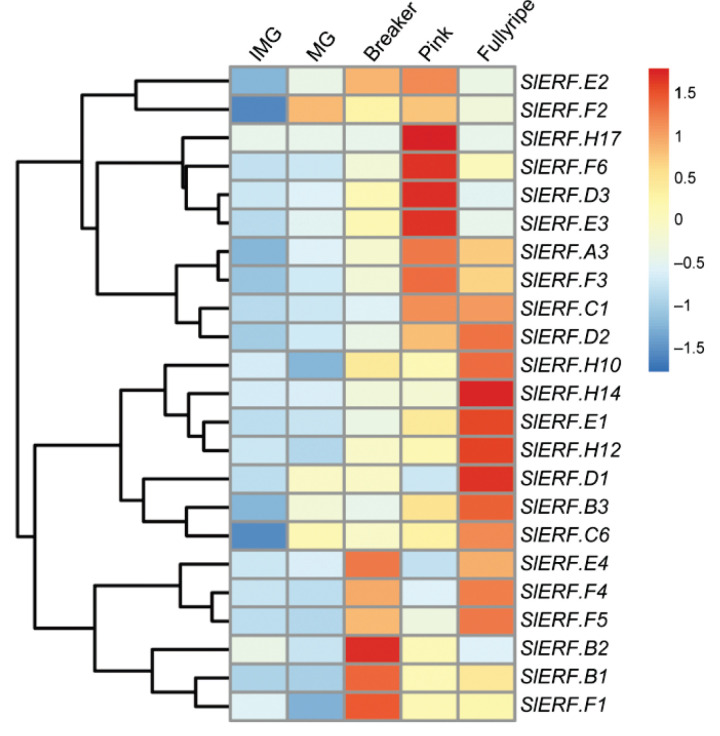
Gene expression pattern of *ERFs* in tomato fruit during ripening. Data for wildtype (*Ailsa Craig*, AC) tomato fruit, downloaded from TomExpress (http://tomexpress.toulouse.inra.fr/, accessed on 15 March 2022 [46]). IMG, immature green fruit, 17 days post-anthesis (dpa); MG, mature green fruit, 39 dpa; Breaker, first sign of color change, 42 dpa; Pink, full color change, 45 dpa; Fully ripe, fruit that have reached peak ripening stage. Genes are clustered by expression patterns using pheatmap package in R 4.2.1 software. The values are displayed in color ranging from blue (low) to red (high), with genes with values below 0.5 excluded from the heatmap.

**Figure 3 cells-11-02484-f003:**
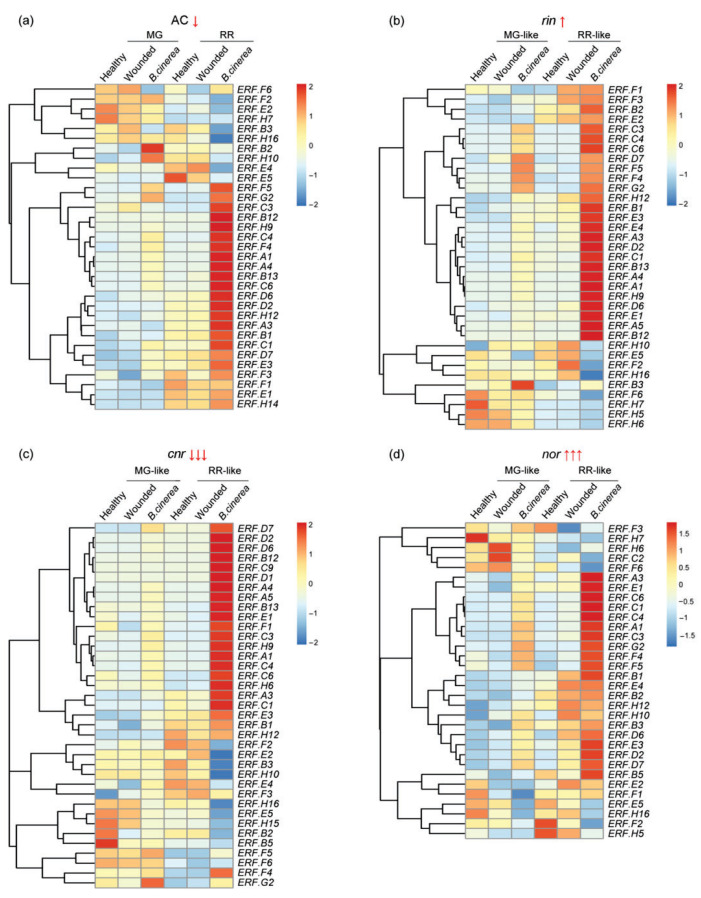
Gene expression pattern of *ERFs* in tomato (**a**) wildtype (*Ailsa Craig*, AC) and ripening mutant (**b**) *rin*, (**c**) *cnr* and (**d**) *nor* in healthy fruit and fruit infected with *B. cinerea* or uninfected (wounded). Wildtype tomato fruits were picked at mature green (MG) and red ripe (RR) stages; the ripening mutant fruits including *Cnr*, *nor*, *rin* were picked at MG-like and RR-like stages, all of which included healthy (healthy, neither wounded nor infected), wounded (wounded but did not infected) and *B. cinerea* (wounded plus infected with *B. cinerea*) fruits. All tomato fruit were sampled at 1 day post-inoculation (dpi) and fruit pericarp and epidermal tissue, excluding seeds, of the blossom end halves of healthy, wounded, and infected fruit were collected. Raw transcriptome data (accession number: GSE148217) were downloaded from the NCBI database and transformed to FPKM using hisat2 tools. The FPKM values were normalized by row and displayed with color ranging from blue (low) to red (high). Genes with FPKM values in all the tissues below 10 were considered as low-level transcripts and are not displayed in the heatmap. A single down arrow sign “↓” indicates medium sensitivity to *B. cinerea* infection, triple down arrow “↓↓↓” indicates extremely sensitive, the up arrow sign of “↑” represents slightly resistant, and the triple up arrow sign “↑↑↑” represents extremely resistant.

**Figure 4 cells-11-02484-f004:**
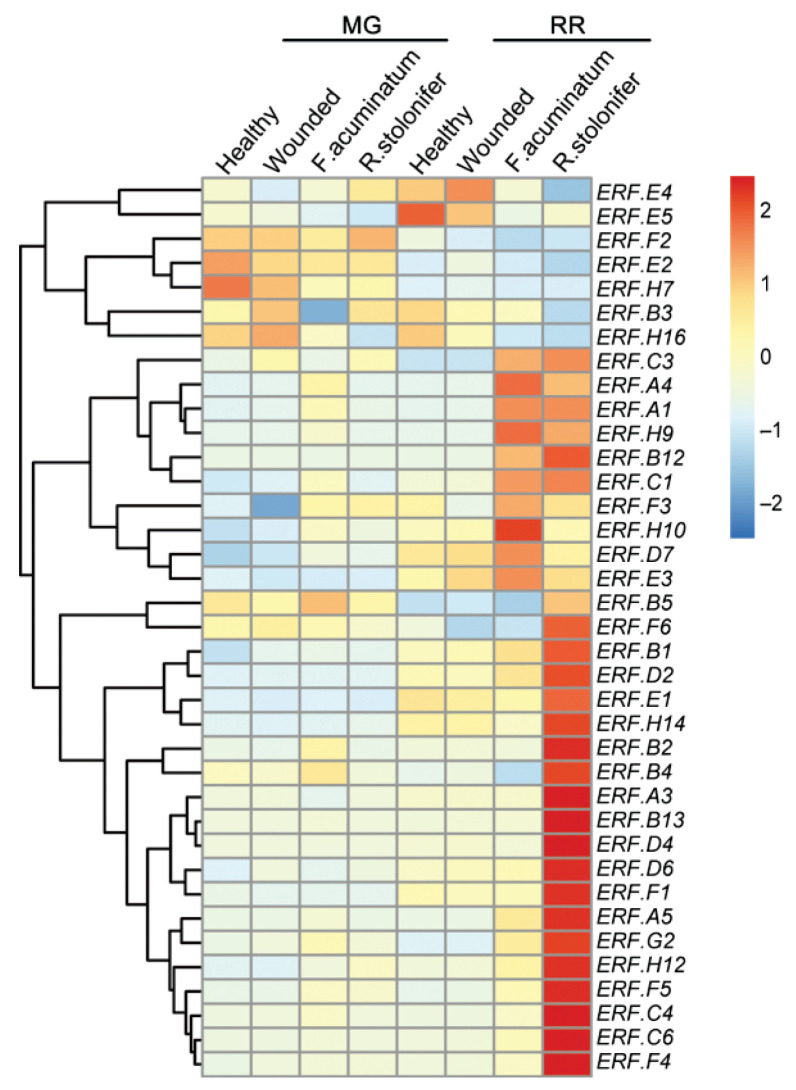
Gene expression pattern of *ERFs* in tomato wildtype (*Ailsa Craig*, AC) fruit infected with *F. acuminatum* and *R. stolonifer.* Wildtype AC tomato fruits were picked at mature green (MG) and red ripe (RR) stages, which included control (wounded but did not infected), wounded plus infected with *F. acuminatum* and *R. stolonifer*, respectively. Raw transcriptome data (accession number: GSE148217) were downloaded from the NCBI database and transformed to FPKM using hisat2 tools. The FPKM values were normalized by row and displayed with color ranging from blue (low) to red (high). Genes with FPKM value below 10 in all the tissues were considered as low-level transcripts and were not displayed in the heatmap.

**Figure 5 cells-11-02484-f005:**
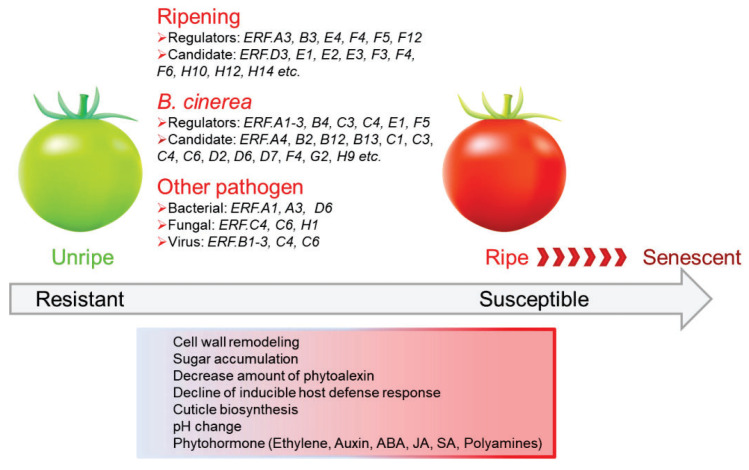
A summary of the changes in regulation of various ERFs in tomato fruit ripening and pathogen response. Increased pathogen susceptibility is an inherent outcome of fruit ripening, which is accompanied by multiple changes, including cell wall degradation and cuticle biosynthesis, decrease of preformed or induced phytoalexin and host defense response reaction, cellular pH change and phytohormone biosynthesis and metabolism. Data from Figure 2, Figure 3, Figure 4 and Table 1.

**Figure 6 cells-11-02484-f006:**
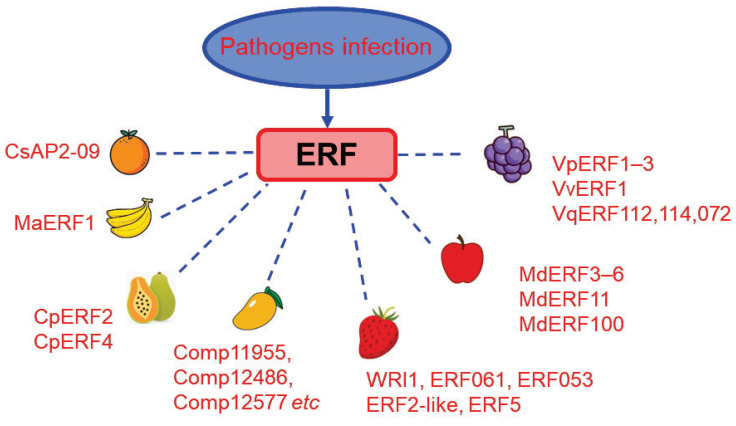
ERF factors functioning in pathogen response in various fleshy fruits. Multiple ERF factors have been identified in different fruits, including citrus (CsAP2-09), banana (MaERF1), papaya (CpERF2, CpERF4), Mango (Comp11955, comp12486, Comp12577 etc.), strawberry (WRI1, ERF061, ERF053, ERF2-like, ERF5), apple (MdERF3–6, MdERF11, MdERF100), grape (VpERF1–3, VvERF1, VqERF112, VqERF114, VqERF072). Further details are described in Section 3.5. Results for tomato are shown in Figure 5.

**Table 1 cells-11-02484-t001:** ERFs involved in pathogenic response of tomato.

ERF	Other Name	Pathogen	Reported Function	Reference
ERF.A1	ERF68	*Pseudomonas syringae* pv. *Tomato* (*Pst*) DC3000; *Xanthomonas euvesicatoria* (*Xeu*); *B. cinerea*	Activation of hypersensitive cell death and disease defense involving modulation of ethylene, SA, jasmonic acid (JA) and hypersensitive response (HR) pathways *ERF.A1* silencing resulted in increased susceptibility to *B. cinerea*, attenuated the *B. cinerea*-induced expression of JA/ethylene-mediated signaling responsive defense genes and promoted the *B. cinerea*-induced H_2_O_2_ accumulation	[36,93]
ERF.A2	ERF1	*B. cinerea*	Expression of *ERF1* was upregulated in fruit after *B. cinerea* infection at MG and RR stage	[86]
ERF.A3	Pti4	*Pst* DC3000; *B. cinerea*;	*ERF.A3* silencing decreased the resistance against *Pst* DC3000, but increased susceptibility to *B. cinerea*; Similar to ERF.A1, *ERF.A3* silencing affected expression of genes involved in JA/ethylene-mediated signaling responsive defense genes and *B. cinerea*-induced H_2_O_2_ accumulation	[36]
ERF.B1		Tomato yellow leaf curly virus (TYLCV)	Transcript of genes were affected in response to TYLCY infection in different cultivated tomato either resistant or susceptible to the virus	[83]
ERF.B2	SlERF5	TYLCV; *B. cinerea*	Similar to *ERF.B1*, the transcript of *ERF.B2* was affected in response to TYLCY infection; *SlERF5* overexpression transgenic tomato plants enhances the resistance to *B. cinerea*	[83,94]
ERF.B3	LeERF4	TYLCV	Similar to *ERF.B1* and *B2*, the transcript was affected in response to TYLCY infection	[83]
ERF.B4		*B. cinerea*	*ERF.B4* silencing increased the susceptibility to *B. cinerea*, which affected expression of genes involved in JA/ethylene-mediated signaling responsive defense genes and *B. cinerea*-induced H_2_O_2_ accumulation	[36]
ERF.C1	*TERF1; JERF2; SlERF1*		ERF1 was involved in chitosan (CHT)-induced systemic acquired resistance (SAR) response	[95]
ERF.C3		*B. cinerea*	*ERF.C3* silencing increased the susceptibility to *B. cinerea*, which affected expression of genes involved in JA/ethylene-mediated signaling responsive defense genes and *B. cinerea*-induced H_2_O_2_ accumulation	[36]
ERF.C4	SlERF01; TSRF1	*Stemphylium lycopersici*; *Ralstonia solanacearum*; *Pst* DC3000; *B. cinerea*; TYLCV	SlERF01 activates expression of *PR1* and plays a key role in SA, JA and ROS signaling pathways, promoting resistance to *S. lycopersici* invasion; A transcriptional activator TSRF1, which was previously demonstrated to regulate plant resistance to *R. solanacearum*, reversely regulates pathogen resistance including *Pst* DC3000 and *B. cinerea;* Similar to *ERF.B1*- *B3*, the transcript was affected in response to TYLCY infection	[83,92,96,97]
ERF.C6	ERF2 or Pti5	*Stemphylium lycopersici*; TYLCV	ERF2 either directly or indirectly regulates *Pto*, *PR1b1* and *PR-P2* expression and enhances tomato resistance to *S. lycopersici*, which has a key role in multiple SA, JA and ROS signaling pathways that contribute to resistance Similar to *ERF.B1*–*B3* and *C4*, the transcript was affected in response to TYLCY infection	[83,98]
ERF.D6	ERF84	*Pst* DC3000	Overexpression of *SlERF84* resulted in decreased plant resistance against *Pst* DC3000, which might due to downregulated expression of *PR* genes.	[99]
ERF.E1	SlERF2	*B. cinerea*	Tomato *ERF2* (*ERF.E1*) participates in MeJA-mediated disease by promoting genes that encode defense enzymes including pathogenesis-related proteins	[35]
ERF.F5	ERF3, SlERF3b	*B. cinerea*	Expression of *ERF3* was upregulated in fruit after *B. cinerea* infection at mature green and redripe stage; *SlERF3b* overexpression transgenic tomato plants enhances the resistance to *B. cinerea*	[86,94]
ERF.H1	SlERF1	*Rhizopus nigricans*	Overexpression of *ERF1* in tomato fruit enhanced resistance against *R. nigricans*, including accumulation of transcripts of *PR1a*, *PR5*, *Chi1* and *PAL* genes	[100]

**Table 2 cells-11-02484-t002:** ERFs involved in pathogenic responses of other fleshy fruits.

Species	ERF	Pathogen	Reported Function	Reference
Strawberry	WRI1, ERF061, ERF053	*B. cinerea*	Upregulated gene expression in *B. cinerea* inoculated fruit	[119]
Strawberry	ERF2-like, ERF5	*B. cinerea*	Upregulated gene expression after *B. cinerea* infection in fruit at Mature-red stage	[120]
Apple	MdERF3, -4, -5, -6	*B. cinerea*	Expression of all four *MdERF* mRNAs is ethylene dependent and also induced by wounding or by *B. cinerea* infection	[121]
Apple	MdERF11	*Botryosphaeria dothidea* (*B. dothidea*)	*MdERF11* overexpression increases the resistance to *B. dothidea* infection, which act through SA synthesis pathway.	[122]
Apple	MdERF100	*Powdery Mildew*	MdERF100 physically interacts with MdbHLH92 which mediates the powdery mildew resistance by regulating the JA and SA signaling pathways	[123]
Grape	VvERF1	*B. cinerea*	Overexpression of *VvERF1* in strawberry fruits reduced the susceptibility to *B. cinerea* infection	[124]
Grape	VqERF112, VqERF114, VqERF072	*Pst* DC3000, *B. cinerea*	VqERF112, VqERF114 and VqERF072 in Chinese wild *Vitis quinquangularis* positively regulate resistance to *Pst* DC3000 and *B. cinerea*	[125]
Grape	VpERF1, VpERF2, VpERF3	*Ralstonia solanacearum*; *Phytophtora parasitica*	VpERF1-3 from a highly powdery mildew (PM)-resistant Chinese wild *Vitis pseudoreticulata*, were positively related to resistance to both bacterial pathogen *Ralstonia solanacearum* and fungal pathogen *Phytophtora parasitica* var. *nicotianae* Tucker.	[126]
Citrus	CsAP2-09	*Xanthomonas citri* subsp. (*Xcc*)	*CsAP2-09* overexpression enhanced the resistance to *Xcc*, while its silence decreased the resistance	[127]
Banana	MaERF1	*Colletotrichum musae*	Heat-induced disease resistance in harvested bananas involves up-regulation of *MaERF1* expression	[128]
Papaya	CpERF2, CpERF4	*Colletotrichum gloeosporioides*	Pathogen stress induces strong accumulation of *CpERF2* and *CpERF4* transcripts. Expression of *CpERF2*, *CpERF4* increases more gradually, reaching maximal levels 14 days after inoculation, with ~5- and ~20-fold increases, respectively	[129]
Mango	Comp11955, comp12486, comp12577 *etc.*	*Colletotrichum gloeosporioides*	Expression levels of 13 *ERF* unigenes (1.5- to 85-fold) were up-regulated in infected fruits.	[28]

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
