# Peer review of "Contrasting Roles of Ethylene Response Factors in Pathogen Response and Ripening in Fleshy Fruit"

_cells, 2022, doi:10.3390/cells11162484_

Round 1

Reviewer 1 Report

This review summarizes the role of ERFs during ripening and pathogen response. This manuscript summarizes a lot of information. However, some details should be included, and the discussion of the relationship between pathogen response and ripening mechanism regulated by ERFs should be improved.

GENERAL CONSIDERATIONS
Please, include the scientific name of the species for a better relationship with the name of the genes.
Please, include the full name of the genes, enzymes and ripening stages. It is necessary to clarify some abbreviations in some parts of the main text.

Regarding the relationship between pathogen response and the ripening mechanism regulated by ERFs, the authors suggest that the hydrolases PG, PME, PL and Exp increase the susceptibility to pathogenic fungi (Lines 245-248). ERF has been described as a regulator of the expression of the PG hydrolases https://doi.org/10.1016/j.postharvbio.2022.111919. Some hypotheses about the induction of pectin degradation regulated by ERF and pathogens induced?

Author Response

Response to Reviewer 1 Comments

This review summarizes the role of ERFs during ripening and pathogen response. This manuscript summarizes a lot of information.

Point 1: However, some details should be included, and the discussion of the relationship between pathogen response and ripening mechanism regulated by ERFs should be improved.

Response 1: We added some more details about the ripening regulators and their function in pathogen response, please see line from 549 to 552 on page 15.

Additionally, we do discuss the relationships between pathogen response and ripening mechanisms regulated by ERFs in tomato and this is summarized in figure 5 on page 14.

GENERAL CONSIDERATIONS

Point 2: Please, include the scientific name of the species for a better relationship with the name of the genes.

Response 2: The Latin names of all species mentioned have been added, please see the revisions in line 43, 67-68, 108-110, 197, 202-207, 212, 219, 225, 229, 233, 542, 545-547, 553, 556, 558, 578.

Point 3: Please, include the full name of the genes, enzymes and ripening stages. It is necessary to clarify some abbreviations in some parts of the main text.

Response 3: We have checked all the abbreviations in the text and figure legends, the names of all necessary genes and enzymes have been defined at first use in the current edition. Please see the revisions in line 25, 49, 83-84, 118, 145-147, 166, 176, 210, 279, 303, 322-323, 327, 565-566.

Point 4: Regarding the relationship between pathogen response and the ripening mechanism regulated by ERFs, the authors suggest that the hydrolases PG, PME, PL and Exp increase the susceptibility to pathogenic fungi (Lines 245-248). ERF has been described as a regulator of the expression of the PG hydrolases https://doi.org/10.1016/j.postharvbio.2022.111919. Some hypotheses about the induction of pectin degradation regulated by ERF and pathogens induced?

Response 4: As mentioned in the text “genes encoding proteins involved in cell wall changes, such as expansin (Exp1), pectinmethylesterase (PME), polygalacturonase (PG) and pectate lyase (PL), all contribute to a reduction in firmness of tomato fruit, which increases susceptibility to pathogenic fungi[70,72-73]”, which indicates the relationship between cell wall genes and firmness, and the correlation between fruit softening and infection by fungal pathogens.

Due to the complex interactions of multiple cell wall genes and their roles in fruit firmness changes (as cited by Ref No.70, 72 and 73), and the complex relationship between fruit texture and fungal pathogen response, it is not possible to put forward a clear hypotheses about the mechanistic relationship between pectin degradation and pathogen infection and response.

Our discussion of the relationships between pathogen response and ripening mechanisms regulated by ERFs, contributes to understanding the relationship between ripening changes and responses to pathogens, as mentioned in Response 1.

We have added the suggested article to our text, please see line from 201 to 202 on page 5: “PpERF/ABR1 bind directly to the promoter of cell wall gene PpPG and activate its expression in peach fruit[55]”.

Reviewer 2 Report

This paper reviewed some interesting aspects of ERF in pathogen response and fleshy fruit ripening.

Although the paper is interesting I am wondering how the authors coordinated pathogen response and ripening. If there is any relationship between these it should be discussed properly.

In the title, ERF should be elaborated.

Keywords should not be taken from the title and abstract.

In section 3, the authors mentioned ROS which is interesting. This should be discussed properly.

many of the references are old. Please update them.

Author Response

Response to Reviewer 2 Comments

This paper reviewed some interesting aspects of ERF in pathogen response and fleshy fruit ripening.

Point 1: Although the paper is interesting I am wondering how the authors coordinated pathogen response and ripening. If there is any relationship between these it should be discussed properly.

Response 1: There is no direct relationship between the control of ripening and the response to pathogens, other than the involvement of ERFs, which are the subject of this review. The results indicate that different ERFs are involved in the two processes, as stated in lines 183-241 page 5 to 6 (fruit ripening and quality formation control) and lines 315-344 on page 8 to 9 and lines from 471 to 519 on pages 13 to 14 (pathogen response). Some ERF have dual regulation roles in the two processes, which was discussed and summarized from 549 to 552 on page 15 and figure 5 on page 14

Point 2: In the title, ERF should be elaborated. Keywords should not be taken from the title and abstract.

Response 2: We have removed ERF from the titles and replaced with ETHYLENE RESPONSE FACTORS and the abbreviation is already defined in the Abstract. The updated keywords were adjusted and more summarized, please see page 1.

Point 3: In section 3, the authors mentioned ROS which is interesting. This should be discussed properly.

Response 3: We have added some more information about the role of ROS in section 3, lines from 263 to 266 on page 7. Detailed sentences are “ROS is not only important because of its effects on plant metabolism but also because the increase in ROS that occurs in response to pathogenic fungal initiates the host plant oxidative burst [10].”.

Point 4: many of the references are old. Please update them.

Response 4: The references (No. 35, 82, 107 in previous edition) were removed, the references (No. 57 to 60 in current edition) from lines 206 to 211 on pages 5 to 6 were adjusted. The references (No. 109-110 in previous edition) were replaced by an updated one (No. 108 in current edition) line 440 on page 12.

Anyway, we remained some representative references, which were published several decades ago but reported discoveries related to the topic of this review.

Round 2

Reviewer 2 Report

The revised version is acceptable.